# Does Enterprise Internal Control Improve Environmental Performance—Empirical Evidence from China

Lijuan Tao *, Xiaoju Wei and Wenjing Wang

Department of Accounting, Business School, Qingdao University, Qingdao 266071, China
* Correspondence: taolj@qdu.edu.cn

**Abstract:** Enterprises are key actors in green governance. Many studies have analyzed the factors that affect corporate environmental performance, but the impact of internal control on environmental performance has not been investigated yet. China's innovative internal control policies make this issue more meaningful for research. Unlike the general practices of developed market economy countries or regions which require enterprises to evaluate and disclose the effectiveness of internal control over financial reporting, China's policy focuses on multi-objective internal control. Using the instrumental variables regression method, this paper employs a moderated mediation model to study the relationship between internal control and environmental performance. This paper takes listed companies on the Shanghai and Shenzhen Stock Exchanges from 2013 to 2021 as the sample. Empirical results show that high-quality internal control is conducive to enhancing environmental performance, while the level of enterprise digitalization plays a mediating role in the relationship between the two, and ownership type moderates the effects of internal control on environmental performance. The conclusion indicates that China's internal control policy is of great significance for the green development of enterprises. Our study contributes to the literature on both the factors affecting environmental performance and the economic consequences of internal control. The study findings can be beneficial for managers in corporations, internal control policymakers and environmental regulators.

**Keywords:** corporate environmental performance; enterprise internal control; enterprise digitalization; IV regression; moderated mediation model

---

## 1. Introduction

Improving environmental performance is an urgent issue. China attaches great importance to the implementation of the 2030 Agenda for Sustainable Development. The construction of ecological civilization in China has entered a new stage with a focus on carbon reduction and the promotion of coordinated and efficient pollution reduction. Hence, carbon emission reduction and pollution prevention are currently important environmental issues that need to be urgently addressed. Enterprises are important entities and key actors in green governance. However, promoting the substantive implementation of green governance in enterprises is a key challenge faced by various sectors in China and elsewhere [1].

Although current literature has identified many factors that affect corporate environmental performance (hereinafter referred to as CEP), enterprise internal control (hereinafter referred to as IC) has received little attention. Generally speaking, listed companies in developed market economy countries or regions are required to evaluate, audit and disclose the effectiveness of internal control over financial reporting (ICoFR). While internal control policies in China require listed companies to evaluate, audit and disclose the effectiveness of multi-objective internal control relating to operations, reporting, compliance and strategy realization (In 2006, six ministries and commissions led by the Ministry of Finance (MOF) established the "Committee of Internal Control Norms for Enterprises", and they

jointly issued the "Basic Internal Control Norms for Enterprises" (the "Basic Norms") on 28 June 2008. The "Basic Norms" requires firms to establish, implement and evaluate internal controls. In terms of its core idea and purpose, the "Basic Norms" was called the Chinese version Sarbanes–Oxley Act (CSOX) by the media. To promote the establishment, implementation, evaluation and auditing of internal controls, on 26 April 2010, MOF, the China Securities Regulatory Commission (hereinafter referred to as CSRC), the National Audit Office and the China Banking and Insurance Regulatory Commission jointly issued the Enterprise Internal Control Guidelines, including "Guidelines for Implementation of Enterprise Internal Controls", "Guidelines for Evaluation of Enterprise Internal Controls" and "Guidelines for Auditing of Enterprise Internal Controls". Those guidelines, together with the "Basic Norms", mark the completion of internal control standard system in China (ICSS). Unlike SOX, internal controls of ICSS are multi-objective-orientated to provide reasonable assurance regarding the achievement of objectives in five categories: ① compliance with applicable laws and regulations, ② safeguarding assets against loss, ③ reliability of financial reporting and related information, ④ effectiveness and efficiency of operations, and ⑤ strategy realization), IC, as a type of institutional resources and dynamic capabilities of the enterprise [2,3], should be able to enhance the environmental management capabilities of enterprises. However, this viewpoint has not yet undergone rigorous empirical testing. In addition, since the internal control standard system (ICSS) was completed in 2010, a large amount of literature has studied the economic consequences of IC, but we still know little about the impact of internal control on CEP.

In addition to being innovative in terms of internal control policy, China also leads the world in terms of enterprise digitalization. According to the Ministry of Industry and Information Technology, the proportion of the digital economy in China's GDP rose from 21.6 percent to 39.8 percent in the last decade, with its scale increasing from CNY 11 trillion in 2012 to more than CNY 45 trillion in 2021. The COVID-19 pandemic has granted an increase in the pace of digital transformation of enterprises. In November 2022, Accenture, an internationally renowned consulting company, found that, in 2022, in the face of a complex and changing environment, digital transformation has become a "must" for more and more Chinese enterprises.

Current literature contains the impact of digitalization on environmental performance, but the conclusions are not uniform [4–7]. What role does internal control play in the relationship between digitalization and CEP? The current research has not yet covered it.

In the context of China's innovative internal control policies, in order to fill the literature gap, this paper aims to investigate the effects of IC on CEP. On one hand, this paper attempts to explore new factors that affect environmental performance and contribute to the sustainable development of human society. On the other hand, this paper also attempts to further explore the new economic consequences brought by China's innovative internal control policies and to provide empirical support for evaluating and improving China's internal control policies.

The empirical research in this article draws three conclusions. First, high-quality internal control and high level of digitalization can improve environmental performance. Second, internal control further enhances environmental performance by enhancing the digitalization level of enterprises. Third, in non-state-owned enterprises, internal control has a greater impact on environmental performance than in state-owned enterprises. This study expands the scope of factors affecting CEP by introducing new potential determinants of environmental performance. Stakeholders in environmental policy and internal control policy can benefit from this study's findings. Enterprise managers can also draw inspiration from the conclusions of this study to improve the performance of environmental management.

The paper proceeds as follows. Section 2 provides the literature review. Section 3 presents the hypotheses. Section 4 illustrates the data, variables and models. Section 5 demonstrates the preliminary empirical results and robustness results. Section 6 provides the discussion of the results. Section 7 concludes this paper.

## 2. Related Literature

### 2.1. Factors Affecting Environmental Performance

There is currently a large amount of literature studying the factors that affect CEP. CEP is a reflection of the level of corporate ethics and a result of strategy selections. Both internal and external governance mechanisms can impact environmental performance significantly [8].

Outside the enterprise, tightening regulatory requirements and intensifying political pressures have brought about large improvements in environmental performance [9]. Empirical evidence from China shows that environmental legislation can significantly improve CEP [10]. However, the impact of various environmental regulations on firms is not uniform. Command-based environmental regulations have a positive impact on CEP, while the impact of market-based environmental regulations is relatively weak [11,12]. The implementation of cleaner production standards and carbon market policy initiation can effectively improve CEP [13,14]. Empirical evidence from Mexico shows that environmental compliance is an important factor in improving the environmental performance of small businesses in emerging economies [15]. Except for policies and regulations, the capital market also plays an indirect role in improving CEP. Empirical studies find that capital market opening, bank deregulation, as well as green credit have a positive effect on CEP [16–19].

Many scholars have conducted research on how to promote scientific decision making in companies from a governance perspective to reduce environmental harm [20]. The behavior of shareholders, board of directors and management in a company determines its environmental awareness and behavior; thus, a good level of corporate governance can help improve CEP. A survey shows that organizational and supervisory factors indicated in research on general and environmental innovation had a positive relationship to employees' environmental innovations [21]. A fact-based research shows that all three aspects of governance—ownership, boards and management—play a role in environmental performance [22]. Empirical evidence of China's listed companies shows that multiple large shareholders can improve CEP [23]. Based on the diversity theory, which posits that diversity improves the quality of management decisions and business ethics, empirical evidence from the US shows that boards' genetic diversity leads to improved environmental performance [24]. Empirical studies also show that a higher proportion of outside board directors [25], a stronger corporate social responsibility (CSR) of the board [26] and the internal heterogeneity of stakeholder groups [27] are associated with more favorable CEP. Analyses of Czech firms and China firms provide evidence that increased state ownership improves CEP [28,29].

The currently available literature have explored the impact of digitalization on environmental performance, but the conclusions are not uniform. Empirical evidence of Chinese pollution-intensive firms [4], U.S. Standard and Poor's 500 companies [6] and large Saudi manufacturing corporations [7] shows a positive relationship between digitalization and environmental performance. Meanwhile, empirical evidence of the Chinese traditional high-polluting textile and apparel industry shows a U-shaped relationship between firm digitalization and environmental performance [5].

### 2.2. The Economic Consequences of Internal Control

There is a large amount of international literature studying the economic consequences of the SOX Act, which requires the ICoFR of listed companies in the US. ICoFR is designed to provide reasonable assurance regarding the achievement of reliability of financial reporting. Thus, the literature in this area usually focuses on the effect of ICoFR on earnings management, capital cost and financial performance such as accrual quality [30], cost of equity [31–33] and stock prices [32].

Internal control policy in China focuses on multi-objectives in several categories relating to operations, reporting, compliance and strategy realization. Therefore, research on the economic consequences of internal control can be described as a hundred flowers

blooming. Except for the effect of IC on corporate financial performance [34,35], the effect of IC on innovation performance has also been widely analyzed [36–39]. In addition, internal control plays a positive role in the strategic formulation process of enterprises, thus it affects corporate strategic choices and strategic performance [40–42].

We can see that the factors that affect CEP and the economic consequences of internal control have always been hot topics in the literature, providing rich results for related research. However, there are still some research gaps left. How does internal control affect CEP? What are the economic consequences of internal control in terms of environmental performance? What are the relationships among IC, digitalization and environmental performance? Does IC have different effects on CEP in enterprises with different types of ownership? These questions have not been answered yet and this paper aims to study these questions to fill the literature gaps.

## 3. Hypothesis Development

### 3.1. Internal Control and Environmental Performance

Based on the objectives of IC and the basic theories of risk management and institutional economics, we argue that high-quality internal control is beneficial for improving CEP.

Firstly, one of the objectives of internal control is to ensure a company's compliance with applicable laws and regulations [43]. In order to promote sustainable development, various countries have formulated corresponding environmental laws and regulations. The implementation of internal control is conducive to these regulations being effectively followed by enterprises, thereby improving their environmental performance.

Secondly, as a tool for risk management [44], internal control can prevent and contain potential risk behaviors that may be detrimental to the overall development of the enterprise through risk assessment and risk control. A sound internal control system helps enterprises to identify and analyze potential environmental risks in a timely manner, enabling them to prevent pollution and develop reasonable environmental risk emergency plans by renovating production equipment, monitoring pollution sources, recycling or utilizing waste before the occurrence of polluting activities. During the process of polluting activities, the internal control system ensures that the emission of pollutants is actively monitored and the amount of emissions is under control. After the occurrence of polluting activities, the internal control system ensures that emergency response mechanisms are activated to reduce the impact of pollution and improve the environmental performance of enterprises.

Thirdly, the essence of internal control determines that companies with a higher quality of internal controls can achieve better environmental performance. From the perspectives of institutional economics and management, internal control is the institutional foundation of corporate governance and the core system of enterprise management, which is of great value in suppressing agency costs in the decision-making process and promoting the effective execution of correct decisions [45].

Existing research indicates that several management factors (such as environmental strategy [46,47] and technological/green innovation [46–50]) are positively related to CEP. IC provides support for these factors. For example, the superordinate objective of internal control is to reasonably ensure the realization of the enterprise's development strategy [51], including environmental strategy. In addition, evidence from China suggests that IC promotes innovation performance through two paths: curbing the opportunism behavior of management and employees, and promoting the communication and transmission of innovation information [52].

Empirical studies found that corporate risk-taking is negatively associated with environmental performance [53]. As the internal risk control mechanism of the organization, internal control has a significant negative impact on corporate risk-taking. This viewpoint has been confirmed by data from Chinese listed companies [54]. Given that internal control can reduce corporate risk-taking, it ultimately benefits CEP.

To sum up, we propose the following hypothesis:

**H1.** *High-quality internal control is conducive for enterprises to improve environmental performance.*

### 3.2. Digitalization and Environmental Performance

Digital technologies such as big data, cloud computing, the Internet of Things and artificial intelligence can play an important role in cleaner production, recycling, energy conservation, emission reduction and carbon reduction [55]. For example, big data, AI, 5G, digital twin technology and other technologies can be used for carbon footprint monitoring, carbon data analysis and intelligent decision making for carbon reduction. Digital technologies provide strong technical support and feasible paths for enterprises to fulfill their environmental responsibilities.

The relationship between digitalization and CEP can be analyzed based on information processing theory [56,57]. The theory highlights the effects of an organization's information and informational processing capabilities on its performance [57]. Digitalization is beneficial for enterprises to collect and integrate data and information in the production and operation process. These data can be used on a real-time basis to obtain prescriptions on future decision making [58]. Once big data analytics are established, decision-making processes will be supported by data-based insights [59]. Environmental management activities are considered to be information-intensive [60,61]. Information sharing helps make better production decisions to reduce pollutant emissions and bring environmental performance [60].

Therefore, we propose the following hypothesis:

**H2.** *Digitalization level is positively correlated with environmental performance.*

### 3.3. Internal Control, Digitalization and Environmental Performance

Digital investment involves enterprises' digital transformation strategy, which is one of the key challenges facing contemporary businesses. The need to leverage digital technology to develop and implement new business models forces firms to reevaluate existing capabilities, structures and culture in order to identify what technologies are relevant and how they will be enacted in organizational processes and business offerings [62]. Digitalization requires enterprises to have the ability to adjust strategies and match resources in response to changes in the internal and external environment. Enterprises need huge internal structural flexibility and external organizational flexibility to emphasize the update or transformation of existing structures, processes and systems [63]. At this point, the role of dynamic capabilities is highlighted.

Based on the resource-based theory and dynamic capabilities theory [2,64], internal control is identified as a type of dynamic capability of an enterprise [65]. Dynamic capabilities refer to the ability to sense opportunities and threats, to seize opportunities and to manage threats and changes [64]. In the internal control framework, control activities are developed based on risk assessment. During the risk assessment process, enterprises should pay attention to factors such as human resources, management, independent innovation and finance internally. Externally, attention should be paid to economic, social and scientific and technological factors such as the economic situation, industrial policies, market competition, consumer behavior and technological progress. These factors are exactly what enterprises with dynamic capabilities are concerned about. Focusing on these factors is beneficial for improving the perception of opportunities and threats, the ability to seize opportunities and the ability to manage threats or changes. Therefore, internal control can be considered as a type of dynamic capability [65].

As a type of dynamic capability, internal control is conducive to the formation and improvement of an organization's "adaptive learning" ability [66], enabling managers to break free from the constraints of habitual thinking and short-sighted thinking, and form a positive and innovative thinking mode [67]. In the process of developing a digital transformation strategy, high-quality internal control can correctly distinguish and coordinate the "skill system" and "convention system", thereby promoting the positive evolution of an organization [66]. Thus, internal control helps enterprises make decisions toward

digitalization and to increase investment. This results in a higher digitalization level, which will improve environmental performance.

The above discussion gives rise to the following hypothesis:

**H3.** *Digitalization plays a mediating role in the relationship between internal control and environmental performance.*

### 3.4. Internal Control, Ownership Type and Environmental Performance

China's listed companies can be categorized into two groups in terms of ownership type: state ownership and private ownership. State-owned enterprises (SOEs) include all companies such that the government has ultimate control. Private enterprises are controlled by a group of shareholders, which includes both private firms and individuals.

Based on the corporate governance theory [68], SOEs are viewed as multitask takers rather than profit maximizers [69]. Therefore, SOEs tend to pursue stability and social welfare in their management modes, operational mechanisms and other aspects [70,71]. In addition, SOEs are closely monitored by the government [68]. As a result, SOEs usually outperform in environmental performance compared to private enterprises to meet their shareholders', i.e., governments' interests and to achieve social welfare [29]. Moreover, SOEs face less competitive pressure due to resource priorities such as policy support, government subsidies and credit financing [72]. This is beneficial for SOEs to make environmental investments and improve their environmental performance [73].

SOEs have stronger political correlations, with their managers deputed by the government. The managers' career prospects are closely related to the extent to which their management executes instructions from their supervisors [29,68]. The improvement of environmental performance does not strongly depend on the effectiveness of internal control, but more on the execution of political directives and other reasons mentioned above. Most private enterprises, on the other hand, are self-reliant and more proactive in utilizing the improvement of internal control to achieve compliance with environmental regulations and improve environmental performance [74]. This strengthens the role of internal control in improving environmental performance. Thus, we hypothesize the following:

**H4.** *Compared to SOEs, the internal control of private enterprises has a stronger positive effect on environmental performance.*

## 4. Data, Variables and Models

### 4.1. Sample Selection and Data Sources

The research topic of this article is proposed in the context of China's innovative internal control policies. In 2012, MOF and CSRC jointly issued the "Announcement on the Implementation of ICSS for Main Board Listed Companies", marking the transition of China's internal control regulatory policies from the "induced change stage" to the "mandatory change stage" [75]. In the same year, the government issued the Development Plan for National Strategic Emerging Industries during the 12th Five-year Plan Period, marking the beginning of China's digital economy policy. Therefore, from the perspective of national policy semantics [76], a sample of listed companies on the Shanghai and Shenzhen Stock Exchange from 2012 to 2021 should be selected. However, considering the use of data with an internal control lag of one year in the model, we trimmed the sample data to 2013–2021. After removing the companies with incomplete data and material internal control weaknesses, we finally obtained an unbalanced short panel data with a total of 23,982 observations of 3666 sample companies spanning 9 years.

The relevant data mainly came from China Stock Market & Accounting Research Database (CSMAR), a comprehensive research-oriented database focusing on China Finance and Economy. The annual reports were downloaded from CNINFO, an official channel for the information disclosure of Chinese listed companies. Environmental performance data came from a database provided by Wind, a leading provider of financial information

services in China. The internal control data came from the Internal Control & Risk Management Database provided by Dibo Company, a leading provider of regulatory technology solutions in China.

We used Stata 17.0 for statistical analysis.

### *4.2. Variables and Measures*

#### 4.2.1. Dependent Variable: Environmental Performance (CEP)

Drawing on previous literature [77], we selected the environmental score of the Huazheng ESG index system to measure CEP. The score is a combination of five aspects of performance: environmental management system, external environment certification, green operating objectives, green products and violations of environmental regulations. The importance of each indicator was evaluated, and different weights were assigned to different indicators. CEP is the weighted comprehensive score of various environmental indicators.

In the robustness test part, we used another index to measure CEP. The data for the new dependent variable came from the CESG database of the CNRDS platform (Chinese Research Data Services Platform). CEP was measured by the scores of positive indicators minus the scores of negative indicators. Negative indicators include environmental penalties and pollution. Positive indicators include environment-friendly products, measures to reduce the three wastes, circular economy, energy conservation, green office, environmental certification, environmental honors and other advantages.

#### 4.2.2. Independent Variable: Internal Control Quality (IC)

When studying the internal control of public firms in China, most of the literature adopts the internal control index provided by Dibo Company, and some literature uses the internal control index developed by H. Chen et al. [78]. Considering the availability of data, we used the Dibo index to measure internal control quality. The Dibo index is calculated through a goal oriented index system [79]. This indicator system consists of indicators at the basic level, operational level and strategic level. The basic level includes three major categories of internal control objectives: reasonable assurance of corporate legality and compliance, asset safety and reliable reporting. The objective of internal control at the operational level is to improve the efficiency and effectiveness of enterprise operations. The objective of internal control at the strategic level is to achieve the development strategy of the enterprise and to achieve sustainable development. Dibo uses a combination of subjective and objective methods to determine indicator weights and determines the final score of the internal control index based on these weights.

#### 4.2.3. Mediating Variable: Enterprise Digitalization (DTF)

Researchers are increasingly relying on automated textual analysis to extract information from corporate disclosures. A particularly popular method is counting word occurrences from word lists that share common meanings [80]. Following the literature [5,81], this paper uses Python crawler technology to obtain the number of digital related words in the annual report. By dividing this number by the total number of words in the annual report, we constructed a relative word frequency as a proxy indicator for digitalization level.

#### 4.2.4. Moderating Variable: Ownership Type

Ownership type (SOE) is a dummy variable measured as 1 if the firm is a state-owned enterprise and 0 otherwise.

#### 4.2.5. Control Variables

On the basis of the existing literature [5,10], we selected the following control variables: (1) Financial leverage (LEV), measured by the ratio of a company's total liabilities to total assets. Environmental investment requires funds, so it may be affected by the company's debt ratio. (2) Profitability (ROA), measured by the ratio of net income to average total assets. Companies with strong profitability have sufficient financial resources to support

environmental governance [82], thereby affecting CEP. (3) We also introduced a dummy variable SPT to reflect whether the company is in a loss state. SPT refer to ST and PT stocks. ST (special treatment) stocks refer to stocks that have suffered losses for two consecutive years and have been subject to delisting risk warnings by the exchange. If a listed company experiences losses for three consecutive years, its stocks will be suspended from the listing. Shanghai and Shenzhen Stock Exchanges then implement "particular transfer services" for stocks that were temporarily suspended from the listing. (4) The age of a company (Age), measured by years of its establishment. The age of a company can reflect its maturity. Enterprises with different maturity levels adopt different development strategies and attach varying importance to environmental issues. Mature enterprises are more inclined to develop pollution prevention and control measures; therefore, their CEP may be higher. (5) Corporate size (LnA), measured by the logarithm of total assets. The larger the size of an enterprise, the easier it is to be regulated. Hence, large companies, which are more sensitive to their public reputations [83], may have better CEP. (6) Ownership concentration (OCR), measured by the sum of the shareholding ratios of the top 10 major shareholders of the company. (7) Industry. Different industries face different regulations and have different impacts on the environment. Therefore, this article controls for industry variables. (8) Year. At the end of 2019, the COVID-19 epidemic broke out and became a global pandemic. It aggravated the global economic crisis and recession. Moreover, the quarantine policy in China affected the normal production and operation of some enterprises. We believe that these impacts change over time. In addition, people's attitudes towards environmental protection will also change over time. Therefore, the appropriate model is a panel data with time-specific effects. Introducing the annual variable Year into the model can control the time-specific effect.

*4.3. Research Model*

In order to test the direct and indirect effect that IC has on CEP, we employed a moderated mediation model, depicted in Figure 1.

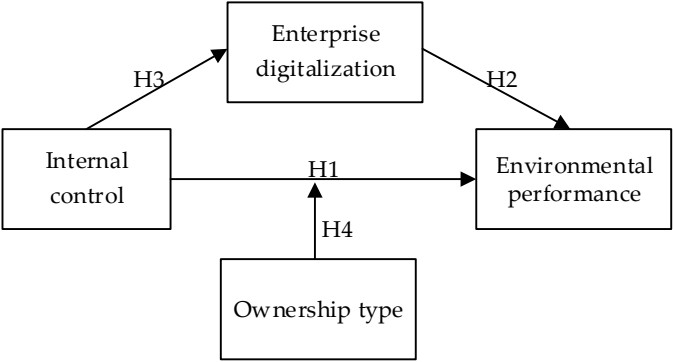

**Figure 1.** Research hypothesis.

The following five models are designed to test the hypotheses.

$$CEP_{it} = \beta_0 + \beta_1 IC_{it} + \beta_2 SOE_{it} + \sum \beta_i Control_{it} + \varepsilon_{it} \tag{1}$$

$$CEP_{it} = \beta_0 + \beta_1 DTF_{it} + \beta_2 SOE_{it} + \sum \beta_i Control_{it} + \varepsilon_{it} \tag{2}$$

$$DTF_{it} = \beta_0 + \beta_1 IC_{it} + \beta_2 SOE_{it} + \sum \beta_i Control_{it} + \varepsilon_{it} \tag{3}$$

$$CEP_{it} = \beta_0 + \beta_1 IC_{it} + \beta_2 DTF_{it} + \beta_3 SOE_{it} + \sum \beta_i Control_{it} + \varepsilon_{it} \tag{4}$$

$$CEP_{it} = \beta_0 + \beta_1 IC_{it} + \beta_2 DTF_{it} + \beta_3 SOE_{it} + \beta_4 SOE_{it} \times IC_{it} + \sum \beta_k Control_{k,it} + \varepsilon_{it} \tag{5}$$

"*Control$_k$*" is a set of control variables. Using these models, we adopted a step-by-step procedure to test for mediation [84–86]. In Models 1 and 4, the coefficient of IC ($\beta_1$) is used to test hypothesis H1. The coefficients of DTF in Model 2 ($\beta_1$) and Model 4 ($\beta_2$) are used to test hypothesis H2. The coefficients of IC in Model 3 ($\beta_1$) and the coefficients of DTF in Model 4 ($\beta_2$) are used to test hypothesis H3. The coefficient of SOE $\times$ IC in Model 5 ($\beta_4$) is used to test hypothesis H4.

## 5. Empirical Results

### 5.1. Descriptive Statistics and Sectoral Information

Descriptive statistics are presented in Table 1. The sample shows an environmental performance with an average of 60.495, with a standard deviation of 8.052, implying a relatively small variance within the sample. The IC variable ranges between 115.470 and 941.310, with an average of 647.110 and a standard deviation of 84.223. The DTF variable ranges between 0 and 353.164, and shows an average value of 11.235, with a high standard deviation (22.057). This indicates that the digitalization level of sample companies varies greatly. The SOE variable shows an average value of 0.366, indicating that 36.6% of the observations are SOEs.

**Table 1.** Descriptive Statistics.

| Variables | Observations | Mean | S.D. | Min | Max |
|---|---|---|---|---|---|
| CEP | 23,982 | 60.495 | 8.052 | 29.460 | 95.160 |
| IC | 23,982 | 647.110 | 84.223 | 115.470 | 941.310 |
| DTF [1] | 23,982 | 11.235 | 22.057 | 0.000 | 353.164 |
| SOE | 23,982 | 0.366 | 0.482 | 0.000 | 1.000 |
| LEV | 23,982 | 0.442 | 0.209 | −0.195 | 1.698 |
| ROA | 23,982 | 0.037 | 0.708 | −2.120 | 108.366 |
| Age | 23,982 | 19.128 | 5.659 | 3.000 | 63.000 |
| LnA | 23,982 | 22.466 | 1.501 | 15.577 | 31.191 |
| OCR | 23,982 | 57.315 | 15.070 | 1.310 | 99.187 |

[1] For the convenience of displaying statistical results, DTF is obtained by multiplying the initial word frequency by 100,000.

Of the 3666 sample companies, 1109 are SOEs and 2426 are from the manufacturing industry classified by the CSRC. Table 2 shows the grouping *t*-test results of state-owned and non-state-owned enterprises. It can be seen that SOEs outperform their privately owned counterparts in environmental performance. This finding is consistent with Wang et al. (2022) [29].

**Table 2.** Two-sample *t*-test.

| CEP | Obs | Mean | S.D. | Min | Max | diff |
|---|---|---|---|---|---|---|
| SOE = 0 | 15,211 | 60.22498 | 0.0655951 | 29.460 | 95.160 | −0.7373855 *** |
| SOE = 1 | 8771 | 60.96237 | 0.0850395 | 31.45 | 95.16 | (−6.8371) |

T statistics in parentheses. * $p < 0.1$, ** $p < 0.05$, *** $p < 0.01$.

Table 3 shows the results of one-way ANOVA when CEP is regarded as the dependent variable and the IND is regarded as a factor. It can be seen that CEP varies significantly across different industries.

**Table 3.** One-way ANOVA with CEP as the dependent variable.

| Source | SS | df | MS | F | Prob > F |
|---|---|---|---|---|---|
| Between groups | 78,593.7593 | 18 | 4366.31996 | 70.88 | 0.0000 |
| Within groups | 1,476,179.01 | 23,963 | 61.6024293 | | |
| Total | 1,554,772.77 | 23,981 | 64.8335254 | | |

*5.2. Baseline Regression Results*

For Model (1), drawing on Filimonova et al.'s (2022) approach [87], we used three econometric models based on panel data: a linear model, a model with fixed effects and a model with random effects. We conducted the F test and Hausman test to identify the best model. In the end, the fixed-effects model was selected. For Models (2)–(5), we performed the same procedures and the fixed-effects models were selected. Table 4 reports the estimation results of these fixed-effects models. To address the impact caused by panel-specific heteroscedasticity, autocorrelation and heteroscedasticity, all the estimates in this paper are provided with robust standard errors.

**Table 4.** Preliminary regression results of Models (1)–(5): fixed-effects panel data model.

| Variables | (1) CEP | (2) CEP | (3) DTF | (4) CEP | (5) CEP |
|---|---|---|---|---|---|
| IC | 0.0014 *** | | 0.0023 ** | 0.0012 *** | 0.0015 ** |
| | (3.0648) | | (2.0308) | (2.6270) | (2.3954) |
| DTF | | 0.0126 *** | | 0.0125 *** | 0.0124 *** |
| | | (3.1151) | | (3.0878) | (3.0847) |
| SOE | 0.6977 ** | 0.5137 * | −0.8320 | 0.5267 * | 0.5119 * |
| | (2.3203) | (1.7200) | (−0.6531) | (1.7584) | (1.7076) |
| SOE × IC | | | | | −0.0006 |
| | | | | | (−0.6510) |
| LEV | 0.4904 | −0.1573 | 0.2360 | −0.0799 | −0.0765 |
| | (1.0683) | (−0.3431) | (0.1672) | (−0.1739) | (−0.1664) |
| ROA | −0.0945 *** | −0.0961 *** | 0.1372 *** | −0.0984 *** | −0.0985 *** |
| | (−6.0356) | (−6.1180) | (3.4555) | (−6.4321) | (−6.4544) |
| Age | 0.2811 *** | 0.0987 *** | 1.9952 *** | 0.1052 *** | 0.1055 *** |
| | (10.7710) | (3.6751) | (28.3485) | (3.9107) | (3.9185) |
| LnA | 0.3839 *** | 0.6171 *** | 3.0808 *** | 0.5921 *** | 0.5917 *** |
| | (2.7209) | (4.3175) | (7.4262) | (4.1259) | (4.1231) |
| OCR | −0.0136 ** | −0.0145 ** | −0.1090 *** | −0.0152 ** | −0.0153 ** |
| | (−1.9698) | (−2.0870) | (−4.8467) | (−2.1872) | (−2.2051) |
| _cons | 47.3835 *** | 47.2696 *** | −88.8288 *** | 46.8794 *** | 46.7330 *** |
| | (12.9370) | (12.8027) | (−10.7955) | (12.7379) | (12.6599) |
| IND | Yes | Yes | Yes | Yes | Yes |
| Year | Yes | Yes | Yes | Yes | Yes |
| N | 23,982 | 23,982 | 23,982 | 23,982 | 23,982 |
| r2_w | 0.0418 | 0.0612 | 0.2184 | 0.0616 | 0.0617 |

T statistics in parentheses, computed using robust standard errors. * $p < 0.1$, ** $p < 0.05$, *** $p < 0.01$.

The coefficient of IC in Model (1) is 0.0014, which is significantly greater than 0. This indicates that, for every 100 points increase in internal control, the average environmental performance increases by about 0.14 points. The higher the level of internal control, the higher the environmental performance. Hypothesis 1 is supported.

The coefficient of DTF in Model (2) is 0.0126, which is significantly greater than 0. This indicates that, for every 100 points increase in digitalization level, the average environmental performance increases by about 1.26 points. The higher the level of digitalization, the higher the environmental performance. Hypothesis 2 is supported.

The coefficient of IC in Model (3) is 0.0023, which is significantly greater than 0. The coefficient of DTF in Model (4) is 0.0124, which is also significantly greater than 0. This indicates that the indirect effect of internal control on environmental performance is significant. Hypothesis H3 is supported. Meanwhile, the coefficient of IC in Model (4), 0.0012, is still significantly greater than 0, indicating a significant direct effect of internal control on environmental performance. The above three coefficients are all greater than 0, providing evidence of "partial mediation" [86]. The proportion of indirect effect in the total effect is $0.0023 \times 0.0124/0.0014 \approx 2.037\%$.

We conducted further tests using the Sobel and Bootstrap methods [86]. The results of the Sobel–Goodman mediation tests are shown in Table 5. The Bootstrap results are

shown in Table 6. From Tables 5 and 6, the same conclusion can be drawn that direct effects, indirect effects and total effects of IC on CEP are significant. Hypothesis H3 is supported.

**Table 5.** Sobel–Goodman Mediation Tests.

|  | Coef | Std Err | Z | $p > |Z|$ |
|---|---|---|---|---|
| Sobel | 0.00008748 | 0.00002649 | 3.303 | 0.00095775 |
| Goodman-1 (Aroian) | 0.00008748 | 0.00002677 | 3.268 | 0.00108469 |
| Goodman-2 | 0.00008748 | 0.0000262 | 3.339 | 0.00084119 |
| a coefficient | 0.007731 | 0.001441 | 5.36451 | $8.1 \times 10^{-8}$ |
| b coefficient | 0.011315 | 0.0027 | 4.19107 | 0.000028 |
| Indirect effect | 0.000087 | 0.000026 | 3.30265 | 0.000958 |
| Direct effect | 0.003813 | 0.000603 | 6.32769 | $2.5 \times 10^{-10}$ |
| Total effect | 0.0039 | 0.000602 | 6.47453 | $9.5 \times 10^{-11}$ |
| Proportion of total effect that is mediated: | 0.02243169 | | | |
| Ratio of indirect to direct effect: | 0.02294641 | | | |
| Ratio of total to direct effect: | 1.0229464 | | | |

**Table 6.** Bootstrap results.

|  | Observed Coefficient | Bootstrap Std. Err. | Z | $p > |z|$ |
|---|---|---|---|---|
| ind_eff | 0.0000875 | 0.0000117 | 7.49 | 0.000 |
| dir_eff | 0.0038125 | 0.0005108 | 7.46 | 0.000 |

In Model (5), the coefficient of SOE × IC is less than 0, as expected by hypothesis H4, but it is not statistically significant. Nevertheless, this is only a preliminary estimate.

*5.3. Robust Regression Results*

5.3.1. Instrumental Variable Regression Model

Models (1) and (3)–(5) with IC as the independent variable face two types of endogeneity problems caused by reciprocal causal relationships and proxy bias. Firstly, the Dibo index is constructed in an outcome-orientated way. Some outcome indicators, such as violations of environmental regulations, are included in the evaluation system [79]. This will engender endogeneity caused by reciprocal causal relationships, leading to inevitable correlations between the dependent variable and the independent variable [88]. Secondly, the Dibo index ignores the premise that internal control can only provide reasonable guarantees for the five objectives, which will also engender endogeneity caused by proxy bias. In Model (2), due to the presence of companies that have been promoting digitalization for many years, there may be a reverse impact of digitalization on internal control. The improvement of the digitalization level is conducive to the improvement of internal control, which will also lead to endogeneity problems caused by reciprocal causal relationships.

Endogeneity bias can lead to inconsistent estimates and incorrect inferences, which may provide misleading conclusions and inappropriate theoretical interpretations. Economists often use instrumental variable (IV) methods to deal with endogeneity problems, and the most popular method is two-stage least squares (2SLS) [89]. Therefore, we used IV methods and 2SLS analysis to solve the endogeneity problems. According to signaling theory [90], companies with higher internal control quality are more likely to conduct comprehensive information disclosure. The internal control quality of the previous period was not affected by the environmental performance or the digitalization level of the current period. Enterprises with internal control weaknesses have poor internal control quality. Therefore, we chose the Internal Control Disclosure Index (ICDI), Internal Control with a lag of one period (LIC) and internal control weaknesses (D, dummy variable, D = 1 if the company has any material internal control weakness, otherwise D = 0) as the instrumental variables for internal control [41,91,92].

Taking Model (1) as an example, we adhered to the following procedure for instrumental variable regression analysis: (1) Model selection. According to the aforementioned test methods in this paper, the fixed-effects model is finally determined. (2) Exogeneity test for instrumental variables. Select all instrumental variables and execute the xtivreg2 command to obtain an instrumental variables estimation as shown in the second column of Table 7. Execute the xtoverid command, and the over-identification test of all instruments shows that Chi-sq (2) = 1.510, *P*-val = 0.4699, which cannot reject the null hypothesis that the excluded instruments are valid instruments. (3) Correlation test for instrumental variables. The first stage regression result is shown in the third column of Table 7, which indicates that all three instrumental variables are significantly correlated with IC. The subsequent LM test of redundancy rejects the null hypothesis that any instrument (ICDI/LIC/D) is redundant. (4) Endogeneity test. The endogeneity test of endogenous regressors shows that Chi-sq (1) = 4.810, *P*-val = 0.0283, thus rejecting the null hypothesis that IC is exogenous at the significance level of 5%.

**Table 7.** Instrumental variable regression results.

| Variables | (1) CEP | (1) IC | (3) DTF | (4) CEP | (5) CEP |
|---|---|---|---|---|---|
| IC | 0.0037 *** | | 0.0101 *** | 0.0036 *** | 0.0052 *** |
| | (0.0012) | | (0.0036) | (0.0012) | (0.0019) |
| DTF | | | | 0.0123 *** | 0.0122 *** |
| | | | | (0.0032) | (0.0032) |
| SOE | 0.5332 ** | −6.7243 * | −0.6980 | 0.5425 ** | 0.4380 * |
| | (0.2304) | (3.8768) | (1.0609) | (0.2300) | (0.2337) |
| SOE × IC | | | | | −0.0042 ** |
| | | | | | (0.0020) |
| LEV | 0.0727 | −37.8574 *** | 0.6970 | 0.0657 | 0.0882 |
| | (0.3511) | (6.4737) | (1.0753) | (0.3511) | (0.3538) |
| ROA | −0.1011 *** | 3.0505 * | 0.1221 ** | −0.1026 *** | −0.1034 *** |
| | (0.0165) | (1.8419) | (0.0480) | (0.0165) | (0.0169) |
| Age | 0.1429 *** | −5.1747 *** | 2.0282 *** | 0.1180 *** | 0.1197 *** |
| | (0.0217) | (0.3466) | (0.0663) | (0.0224) | (0.0226) |
| LnA | 0.5802 *** | 11.9972 *** | 2.9231 *** | 0.5438 *** | 0.5414 *** |
| | (0.1011) | (1.7618) | (0.3085) | (0.1013) | (0.1017) |
| OCR | −0.0178 *** | 0.4458 *** | −0.1141 *** | −0.0165 *** | −0.0172 *** |
| | (0.0052) | (0.0812) | (0.0161) | (0.0051) | (0.0052) |
| ICDI | | 3.8950 *** | | | |
| | | (0.1313) | | | |
| LIC | | 0.1490 *** | | | |
| | | (0.0090) | | | |
| D | | −64.2862 *** | | | |
| | | (4.7650) | | | |
| IND | Yes | Yes | Yes | Yes | Yes |
| Year | Yes | Yes | Yes | Yes | Yes |
| N | 23,767 | 23,767 | 23,767 | 23,767 | 23,767 |

T statistics in parentheses, computed using robust standard errors. * $p < 0.1$, ** $p < 0.05$, *** $p < 0.01$.

For Models (3)–(5), we performed similar tests and the instrumental variable model was proven to be necessary (Unlike Model (1), ICDI and LIC are selected as instruments for IC in Model (3)). Table 7 presents the results of the instrumental variable regression models. It can be found that the robust results are basically consistent with the preliminary results, except that the coefficient of SOE × IC is significantly less than 0 and the estimated effect of IC on CEP is stronger than that of the preliminary results. Therefore, all the five hypotheses are verified.

5.3.2. Alternative Method and Dependent Variable

We adopted an alternative method to test hypothesis H5. Based on ownership type, we divided the full sample into two sub-samples, with one sub-sample consisting of SOEs

and the other consisting of private enterprises. After endogeneity testing, the results are shown in Table 8, from which it can be found that IC is positively correlated to CEP for both sub-samples. However, the effect of IC on CEP is significant only for the sub-sample of private enterprises, which implies the moderating role of ownership type on the relationship between internal control and environmental performance.

**Table 8.** Regression results for SOEs and private enterprises.

|  | SOEs | | Private Enterprises | |
|---|---|---|---|---|
| **Variables** | **CEP** | **CEP** | **CEP** | **CEP** |
| IC | 0.0010 | 0.0009 | 0.0042 *** | 0.0041 *** |
|  | (0.0007) | (0.0007) | (0.0014) | (0.0014) |
| DTF |  | 0.0301 *** |  | 0.0087 *** |
|  |  | (0.0101) |  | (0.0029) |
| Control | Yes | Yes | Yes | Yes |
| N | 8771 | 8771 | 14,990 | 14,990 |

Note: t statistics in parentheses, computed using robust standard errors. * $p < 0.1$, ** $p < 0.05$, *** $p < 0.01$. For the SOEs sample, preliminary results of fixed-effects panel data models are listed here because the endogeneity issue is not material. The instrumental variable regression results are listed in the table for the private enterprises sample.

We replaced CEP with the CNRDS index and re-estimated the models. The results are shown in Table 9. It can be found that the re-estimation results using the alternative dependent variable are consistent with the baseline regression results.

**Table 9.** Instrumental variable regression results using data from CNRDS.

| **Variables** | **(1) CEP** | **(2) CEP** | **(3) DTF** | **(4) CEP** | **(5) CEP** |
|---|---|---|---|---|---|
| IC | 0.0003 *** |  | 0.0000 *** | 0.0003 *** | 0.0005 *** |
|  | (0.0001) |  | (0.0000) | (0.0001) | (0.0001) |
| DTF |  | 136.4413 ** |  | 127.5602 ** | 124.7352 ** |
|  |  | (62.5355) |  | (62.5120) | (62.6207) |
| SOE | −0.0912 ** | −0.0977 ** | −0.0000 | −0.0906 ** | 0.1320 |
|  | (0.0391) | (0.0389) | (0.0000) | (0.0391) | (0.0867) |
| SOE × IC |  |  |  |  | −0.0004 *** |
|  |  |  |  |  | (0.0001) |
| LEV | 0.0015 | 0.0009 | 0.0000 *** | 0.0014 | 0.0017 |
|  | (0.0018) | (0.0023) | (0.0000) | (0.0018) | (0.0018) |
| ROA | −0.0192 *** | −0.0174 *** | 0.0000 | −0.0193 *** | −0.0195 *** |
|  | (0.0050) | (0.0051) | (0.0000) | (0.0050) | (0.0050) |
| Age | 0.0673 *** | 0.0596 *** | 0.0000 *** | 0.0647 *** | 0.0652 *** |
|  | (0.0044) | (0.0042) | (0.0000) | (0.0046) | (0.0046) |
| LnA | 0.2670 *** | 0.2818 *** | 0.0000 *** | 0.2633 *** | 0.2635 *** |
|  | (0.0182) | (0.0171) | (0.0000) | (0.0183) | (0.0183) |
| OCR | 0.0003 | 0.0006 | −0.0000 *** | 0.0004 | 0.0003 |
|  | (0.0009) | (0.0009) | (0.0000) | (0.0009) | (0.0009) |
| IND | Yes | Yes | Yes | Yes | Yes |
| Year | Yes | Yes | Yes | Yes | Yes |
| N | 25,750 | 25,750 | 25,750 | 25,750 | 25,750 |

T statistics in parentheses, computed using robust standard errors. * $p < 0.1$, ** $p < 0.05$, *** $p < 0.01$.

### 5.3.3. Empirical Results for a Larger Sample

In the previous text, we selected samples from the perspective of national policy semantics [76]. However, the "Basic Internal Control Norms for Enterprises" was issued on 28 June 2008. Since the release of the "Basic Norms", many listed companies have voluntarily followed the requirements of the "Basic Norms" for comprehensive internal control construction and information disclosure. Therefore, we extended the sample data

to 2009 and beyond, i.e., a larger sample of 2009–2021. Table 10 presents the results of the instrumental variable regression models for the larger sample.

**Table 10.** Instrumental variable regression results for the larger sample.

| Variables | (1) CEP | (2) CEP | (3) DTF | (4) CEP | (5) CEP |
|---|---|---|---|---|---|
| IC | 0.0059 *** | | 0.0091 ** | 0.0058 *** | 0.0096 *** |
| | (0.0013) | | (0.0038) | (0.0013) | (0.0022) |
| DTF | | 0.0201 *** | | 0.0195 *** | 0.0193 *** |
| | | (0.0027) | | (0.0027) | (0.0027) |
| SOE | 0.7747 *** | 0.7570 *** | −1.2893 ** | 0.7993 *** | 0.5466 *** |
| | (0.1949) | (0.1928) | (0.6256) | (0.1936) | (0.2001) |
| SOE × IC | | | | | −0.0089*** |
| | | | | | (0.0021) |
| LEV | 0.4084 ** | 0.1139 | −1.1560 | 0.4300 ** | 0.4567 ** |
| | (0.1911) | (0.1664) | (0.7300) | (0.1943) | (0.1945) |
| ROA | −0.0835 *** | −0.0725 *** | 0.1688 ** | −0.0867 *** | −0.0867 *** |
| | (0.0176) | (0.0219) | (0.0834) | (0.0175) | (0.0175) |
| Age | 0.2279 *** | 0.1798 *** | 0.6242 *** | 0.2156 *** | 0.2084 *** |
| | (0.0184) | (0.0167) | (0.0489) | (0.0184) | (0.0179) |
| LnA | 0.7541 *** | 0.7921 *** | 3.6558 *** | 0.6831 *** | 0.7034 *** |
| | (0.0755) | (0.0710) | (0.2496) | (0.0754) | (0.0743) |
| OCR | −0.0348 *** | −0.0288 *** | −0.1082 *** | −0.0327 *** | −0.0356 *** |
| | (0.0041) | (0.0040) | (0.0131) | (0.0041) | (0.0043) |
| IND | Yes | Yes | Yes | Yes | Yes |
| Year | Yes | Yes | Yes | Yes | Yes |
| N | 30,087 | 30,087 | 30,087 | 30,087 | 30,087 |

T statistics in parentheses, computed using robust standard errors. * $p < 0.1$, ** $p < 0.05$, *** $p < 0.01$.

It can be found that the results of a larger sample are basically consistent with the preliminary results, except that the coefficient of SOE × IC is significantly less than 0 and the estimated effect of IC on CEP is stronger than that of the preliminary results. Therefore, all the five hypotheses are verified.

In conclusion, the robustness results are consistent with the baseline regression results and provide further confidence in the hypothesis.

## 6. Discussion of Findings

In our study, we focus on investigating the effects of internal control on CEP. We divide the effects into direct effects of internal control on CEP and indirect effects mediated by digitalization. In addition, we also study how the nature of ownership can moderate the effects of internal control on CEP.

The empirical results show that there is a significant effect of internal control on CEP (H1). The findings are consistent with the essence of internal control. Within the existing literature, the essence and mechanism of internal control is mainly discussed from three perspectives: external auditing, economics and organization theory [93]. The literature has long deviated from the perspective of traditional auditing, which is mainly concerned with lower level controls related to specific cycles, processes and transactions. From an economics perspective, internal control is considered as an institutional arrangement to reduce agency costs. Agency costs arise in any situation involving cooperative effort. A company is simply one form of legal fiction which serves as a nexus for contracting relationships; hence, there are agency costs generated at every level of the company [94]. In the company, environmental management activities need cooperative efforts among individuals. Internal control provides monitoring and bonding mechanisms that are necessary to improve environmental performance. The result of this paper expands the application scope of agency theory in the field of internal control and environmental management.

The empirical results show that there is a significant indirect effect of internal control on CEP (H3), and that is consistent with what we expected earlier, that is, internal control as

a type of complementary resources and capabilities can not only affect CEP directly, but also indirectly affect CEP by improving digitalization. The literature from an organization theory perspective, or strategic management perspective, considers internal control as a type of organizational resource or a type of dynamic capability. In this area, the resource-based theory or dynamic capability theory is usually referred to [2,64,65]. Specifically, internal control is seen as complementary resources and capabilities because they have limited ability to generate competitive advantage in isolation. However, in combination with other resources and capabilities, it can enable a firm to realize its full potential for competitive advantage [65,95]. We find that digitalization has a positive impact on CEP, while IC improves the digitalization level of the firm and makes a further boost to environmental performance. This result is consistent with the resource-based view and provides new empirical evidence for the theory.

The empirical results support the positive significant effect of Digitalization on CEP (H2), which is consistent with the findings of some previous studies [56,60]. The fourth hypothesis related to the moderating role of ownership type in the relation between internal control and CEP is also supported by the regression results (H4). Compared to SOEs, the internal control of private enterprises has a stronger positive effect on environmental performance. This finding is consistent with the literature in the corporate governance field [68,69] but somewhat contrary to the findings of Zhang and Zhao (2022) [73].

In short, the study findings contribute to the literature on both the factors affecting environmental performance and the economic consequences of internal control. On one hand, this study has found a new factor that affects environmental performance. On the other hand, previous literature has studied the positive effect of IC on firm's financial performance [34,96], while the research findings of this article provide additional evidence on the economic consequences of internal control in terms of environmental performance.

## 7. Conclusions, Implications, Limitations and Further Research Directions

In this paper, we study the effect of internal control on environmental performance. This is a topic that has not yet been explored in previous literature. Using a moderated mediation model and the data of Chinese listed companies, we find that internal control can help improve environmental performance. Internal control's causal effect can be apportioned into its indirect effect on CEP through enterprise digitalization and its direct effect on CEP. Compared to SOEs, the internal control of private enterprises has a stronger positive effect on environmental performance.

From the perspective of practice, this study serves to provide beneficial reference for managers of the firms as well as policymakers of the governments with the aim of promoting environmental performance and the sustainable development of human society. Nowadays, environmental protection issues brook no delay. As the largest developing country in the world, China will pursue green development by promoting a green and low-carbon development model and lifestyle, actively addressing the climate change and protecting ecological system. This macro environment provides development goals and external governance backgrounds for the micro individual enterprise. So how can enterprises achieve the goals of external governance and green development? This article provides a feasible answer from the perspective of internal control. By establishing sound internal control systems, and effectively implementing internal controls, enterprises can have better environmental performance.

Since the establishment of a complete ICSS in China in 2010, the government has been actively encouraging enterprises to implement it. Departments such as MOF and CSRC have issued several announcements, notifications and other documents requiring listed companies to implement ICSS in batches. SOEs controlled by central government and local governments are required to fully implement ICSS by 2012. Private companies listed on the main board with a total market value of more than CNY 5 billion as of 31 December 2011 and an average net income of more than CNY 30 million over the 2009–2011 period, should implement ICSS by 2013. Other main board listed companies should implement ICSS by

2014. The results found in this paper reveal that the establishment and implementation of internal control not only improves efficiency and effectiveness of operations, compliance with applicable laws and regulations, but also contributes to the sustainable development of the enterprise, establishing a reputation for fulfilling social responsibilities. This marginal effect of internal control provides managers with more incentives to implement ICSS actively. More importantly, in private enterprises, internal control has a stronger effect on CEP than in SOEs. Therefore, private enterprises should pay more attention to internal control policies and actively follow internal control norms.

The research findings of this article have strong policy implications. Based on these findings, the study proposes several policy recommendations. (1) On one hand, policies that include environmental protection goals are needed; on the other hand, it is also necessary to clarify specific details on how to implement environmental policies. The policymakers or other professional institutions can establish internal control standards aimed at environmental protection or add environmental elements to the existing categories of internal control objectives. Taking the globally renowned COSO framework as an example, the objective "effectiveness and efficiency of operations" can be expanded to "effectiveness, efficiency and sustainability of operations". (2) We suggest that Chinese policymakers and regulatory agencies should actively strengthen the improvement and implementation of internal control policies. In China, the internal control standard system is relatively complete. However, the "Guidelines for Implementation of Enterprise Internal Controls" can still be further modified to provide additional guidance aimed at environmental protection. (3) Our researching findings will also bring better insights and illumination to the policymakers of other countries. We suggest that policymakers and regulatory agencies in other countries draw on China's practices and expand the scope of internal control policy to include objectives other than reliable reporting. After the introduction of China's innovative internal control policies, two conflicting views emerged in the capital market. One viewpoint holds that the system innovation has really caught the concerns of top managers and expressed the role and effectiveness of internal control to the greatest extent. Some international experts even consider it as the most "positive and effective" response to the international financial crisis in 2008 [43]. However, considering the critical and controversial history of the SOX Act in the United States, another viewpoint holds that China's comprehensive disclosure policy is too radical. From an empirical point of view, is this response really "positive and effective"? The results are uplifting as the conclusion confirms to a certain extent that China's "radical" or "innovative" comprehensive disclosure policy is "positive and effective".

It should be noted that this article has the following limitations. Firstly, the sample only includes listed companies, which usually outperform non-listed companies in size and profitability. Thus, the conclusions and suggestions drawn in this paper cannot be directly generalized to non-listed enterprises or other small and medium-sized enterprises, which are also key participants of green governance. Requiring all enterprises to improve their internal controls in order to improve environmental performance may not be in line with the principle of cost-effectiveness. After all, the construction and implementation of internal control will bring costs to the enterprise. Secondly, the effects of internal control are different from that of internal control policies. The research conclusion of this article only provides indirect evidence for the effects of China's internal control policies.

What are the causal effects of internal control policies on CEP? This is a meaningful question worth studying in the future. However, the benefits of policies are, by their nature, difficult to isolate. Internal control policies were formulated amidst sharp financial, economic and environmental changes. It makes a large number of simultaneous, disparate policy changes, which continue to be implemented and phased in over time. The implementation of the policies is also being accompanied by a host of other, overlapping capital market policy changes. From the perspective of econometrics, randomized experiments can be used as gold standards for causal inference [97–99], but experiments are time-consuming, expensive and may not always be practical [97]. Instead, quasi-experimental research de-

signs which employ the logic of experimentation are being developed by researchers. Therefore, we suggest that quasi-natural experimental methods, such as the difference-in-differences method or the regression discontinuity method, be used to directly study the causal effects of internal control policies on CEP. If the treatment effect of internal control policies on CEP is significant, then it is more reasonable to suggest that other countries learn from China's policies.

**Author Contributions:** Conceptualization, L.T.; methodology, L.T., W.W. and X.W.; software, L.T.; resources, W.W.; validation, L.T., W.W. and X.W.; formal analysis, L.T.; investigation, X.W., L.T. and W.W.; writing—original draft preparation, L.T.; writing—review and editing, X.W.; supervision, L.T.; project administration, L.T.; funding acquisition, L.T. All authors have read and agreed to the published version of the manuscript.

**Funding:** This research was funded by Natural Science Foundation of Shandong Province (grant number ZR2022MG067).

**Institutional Review Board Statement:** Not applicable.

**Informed Consent Statement:** Not applicable.

**Data Availability Statement:** Not applicable.

**Acknowledgments:** We would like to thank the editor and anonymous referees for their constructive comments that helped to improve the paper.

**Conflicts of Interest:** The authors declare no conflict of interest.

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
