# Peer review of "Does Enterprise Internal Control Improve Environmental Performance—Empirical Evidence from China"

_sustainability, doi:10.3390/su151310199_

Round 1

Reviewer 1 Report (Previous Reviewer 2)

Thank you for the amendments.

Author Response

Dear Reviewer,

On behalf of my co-authors, we thank you very much for your nice comments on our manuscript entitled “Does Enterprise Internal Control Improve Environmental Performance——Empirical Evidence from China” (ID: sustainability-2447986).

Your comments and suggetions are all valuable and very helpful for improving our paper. Some of the cited references are not relevant to internal control or environmental performance, but they are relevant to research methods. Hence, we have introduced some seemingly unrelated references in the final section.

Once again, thank you very much for your comments and suggestions.

Reviewer 2 Report (Previous Reviewer 3)

The authors sufficiently addressed the crticis. I suggest that discussion and conclusion should be separate sections. Discussion should be come after empirical analyses and then Conclusion. Conclusion should include limitations, key findings, policy suggestions, and further research directions

There are some minor typos and it should be checked one more time.

Author Response

Reviewer 3 Report (New Reviewer)

Title: “Does Enterprise Internal Control Improve Environmental Performance - Empirical Evidence from China”

The article is prepared on a current topic; the article is of interest to scientists; and the article can be published, but it needs preliminary thorough revision. The article is devoted to the study of effects of internal control on environmental performance.

 Notes and recommendations for authors:

1.      In my opinion, it is worth specifying the research question at the beginning of the article

2.      In my opinion, the title of the article should be changed

3.      In the text of the article, there are fragments of the text highlighted in red - it is not clear why the authors did this

4.      Lines 44, 60 - [Error! Reference source not found.,2]

5.      In the "Introduction" section, the authors write a lot about the literature review, but, in my opinion, they pay too little attention to revealing the relevance of the research topic

6.      I consider it appropriate to divide the article into separate sections "Discussion" and "Conclusions"

7.      In the section Conclusions to determine how these hypotheses were proven within the study

8.      The Discussion section should be developed by supplementing it with links to specific studies and publications by scientists who have studied similar research problems

9.      At the end of the article, it is worth indicating the limitations of the study

Author Response

This manuscript is a resubmission of an earlier submission. The following is a list of the peer review reports and author responses from that submission.

Round 1

Reviewer 1 Report

I read the manuscript “Dose Enterprise Internal Control Improve Environmental Performance——Empirical Evidence from China” with great interest. I have some serious concerns that need to address before the consideration of this manuscript for publication.

1. Abstract did not contain any information about the data set.

2. Introduction is unnecessarily long and does not convey the exact message.

3. Literature gap is not clearly presented, and the reason for conducting this study is not demonstrated clearly.

4. Clearly state your research question; why and what are you studying in this manuscript?

5. Is there any theoretical support for your proposed framework? No information is available in the manuscript.

6. Why are only A-share markets from Shanghai and Shenzhen Stock Exchange from 2013 to 2021 selected as the sample? Need logical justification and rationale.

7. How are study variables computed? There is no information.

8. The information provided to select the econometric model “For model (1), we first estimate a pooled regression model. Then, a time and entity fixed- effects model is estimated. F-test rejects pooled regression model (F=14.47, Prob>F=  0.0000). After that, a time and entity random - effects model is estimated. Hausman test  rejects the random- effects model (chi2 (31) = 254.64, Prob > chi2 = 0.0000). In the end, the fixed-effects model is selected. For model (2) and model (3), we perform the same proce- dures and the fixed-effects models are selected.” Is miss leading and does not convey proper information.

9. Rationale behind selecting 2sls Econometric model is not provided. Why is it sufficient for this study?

10. How the issue of endogeneity, panel-specific heteroskedasticity, autocorrelation, and heteroscedasticity are addressed in this manuscript? There is no information.

11. No policy implication is provided, or the theoretical contribution of this manuscript discussed in detail. Why and how are study findings beneficial for China?

12. Limitation of this study and future work directions needs to be highlighted in depth. 

Some grammatical mistakes need to address. 

Sentence structure and engagement with the audience need to improve.

Reviewer 2 Report

Please kindly attend to the comments in the attachment.

The H3 hypothesis requires some attention. Otherwise, overall, it is a well written and cogent piece of work.

Thank you.

English is good except for some minor amendments.

Reviewer 3 Report

The research topic is interesting and compatible with the Journal, but it needs serious revision.

The study is focused environmental effect of internal control, but the authors should also consider the papers investigating the interplay between governance and environment for improving the novelty, literature review and discussion parts of the study.

The authors should clearly describe their sample (for example sectoral information). The results change among sectors?

The study period covers the Covid-19 as I see, but nothing about the Covid in analysis and discussion parts.

The authors should separate the Discussion from the Conclusion and considerably improve it.

The authors benefited relatively a limited literature. I suggest they review the whole literature and revise the literature, discussion and conclusion sections.

The limitations of the study and future research suggestions should be clearly explained.

The paper includes some typos and grammatical errors.
